# Research

organic chemistry/synthetic chemistry/materials science

precursor, chemical vapour deposition, parylene

**Author for correspondence:**
Daniel Kast
e-mail: kast.daniel@yahoo.de

# Improved route to a diphenoxide-based precursor for chemical vapour deposition of parylene AF-4

Daniel Kast[1], Gerhard Franz[1] and Jay J. Senkevich[2]

[1]Department of Applied Sciences and Mechatronics, Munich University of Applied Sciences, München, Germany
[2]Massachusetts Institute of Technology, Cambridge, MA, USA

DK, 0000-0001-7650-4094; GF, 0000-0003-3823-3697

In this work, we present the synthesis of an alternative precursor for chemical vapour deposition of parylene AF-4 to the widely used standard, octafluoro[2.2]paracyclophane. The standard precursor suffers from uncertainties in its supply chain and its synthesis is of low yield. A comparison between different reaction parameters and solvents is drawn by means of thermal, laboratory-scale and microwave-assisted reactions and quantitative nuclear magnetic resonance (qNMR) studies.

## 1. Introduction

Parylene AF-4 is a high-temperature stable polymer for conformal coatings with a broad field of applications [1–3]. It is produced by chemical vapour deposition (CVD) via the Gorham method [4]. The problem though is a very high precursor cost [5,6]. Materials which are currently used as precursors are shown in figure 1. Octafluoro[2.2]paracyclophane (1), the precursor for parylene AF-4, is 75 times the cost of [2.2]paracyclophane, the precursor for parylene N, with a single-source supplier (Yuan-Shin Materials Technology Corp. Ltd, Kaohsiung, Taiwan). Although progress has been made, the difficulty with this synthesis is that it requires a total of three steps and the last one is not scalable and of low yield [6–8]. An alternative route to parylene AF-4 is to start with one of the intermediate synthetic products 1,4-bis(bromodifluoromethyl)-benzene (2) as the CVD precursor [9]. On this route, the experimental equipment has to withstand the highly reactive bromine radicals, which are very corrosive. Therefore, it is desirable to have a scalable synthetic route to the parylene AF-4 precursor that circumvents the production of corrosive by-products during the cracking process. For detailed information about the CVD of parylene AF-4 with the diphenoxide-based precursor, please see the paper 'Parylene AF-4 via the Trapping of a Phenoxy Leaving

**Figure 1.** Suitable CVD precursors.

**Figure 2.** Synthesis of the diphenoxide precursor proposed by Senkevich [6].

**Figure 3.** Synthetic route to the diphenoxide pursued in the current work.

Group' of Senkevich, which is available upon open access (Research Gate) [6]. In this paper, he proposed a three-step synthesis of 1,4-bis[difluoro(phenoxy)methyl]benzene (3) from diphenylterephthalate as the starting material, as is shown in figure 2. This route harbours the benefit of a late-stage fluorination to omit peculiarities of fluorine chemistry during the synthesis. This ensures a theoretically high yield for all the preceding synthetic steps [10–14]. A schematic representation of the synthetic route of Senkevich is laid out in figure 2. However, due to the practically uncontrollable formation of S-containing side products, the preparation of the thiocarbonyl compound is hardly attainable at all. We present an alternative route towards obtaining the diphenoxide in high purity for its use as a viable precursor for the chemical vapour deposition of parylene AF-4 in one step. 1,4-Bis[difluoro(chloro)methyl]-benzene is treated with sodium phenoxide in DMPU under reflux conditions for one hour to yield the diphenoxide precursor for parylene AF-4 via nucleophilic substitution. The scheme of the reaction is outlined in figure 3.

The microwave experiments have shown that temperature, reaction time, excess of reagent and solvent are the key parameters for achieving optimal yield. When using NMP, even at high temperatures, the yields were quite poor. On the other hand, using DMPU on reflux gave a modest yield of 78%. Earlier protocols like the one of Dolbier *et al.* [3] used highly toxic HMPA. We decided to replace it by DMPU, which poses a far smaller safety hazard and environmental concerns, and is therefore a very important consideration for scale-up [15]. The solvent HMPA is known to favour $S_N2$ reactions and to increase their rate. How exactly the mechanism works is though not clear among all authors of previous related publications [3,16,17]. In microwave reactions and subsequent qNMR studies, the influence of temperature, time and excess molar equivalents of sodium phenolate were compared. The results are listed in table 1.

Apparently, high temperatures above 150°C are needed to achieve a considerable conversion of the dichloride to the diphenoxide. The reaction temperature in the experiments of Dolbier *et al.* [3] and Guidotti *et al.* [16] was between 100°C and 120°C. Therefore, for thermal laboratory scale preparations, DMPU under reflux for one hour seems to be a good choice of parameters. To obtain reasonable yields with NMP, one would have to raise the reaction temperature so high that it would not be feasible for laboratory use and may require special equipment like an autoclave or a microwave reactor [18]. Temperature variations in this regime have little influence on the yield of the desired reaction product—the diphenoxide. A large molar excess of sodium phenolate seems to have a more

**Figure 4.** Formation of parylene AF-4 via a stabilized radical intermediate.

**Table 1.** Compositions of the reaction mixtures after the microwave experiments.

| component | percentage in mixture | | | | | |
|---|---|---|---|---|---|---|
| starting material | 18 | 3 | 0 | 0 | 0 | 0 |
| monophenoxide | 61 | 23 | 14 | 0 | 0 | 0 |
| diphenoxide | 27 | 31 | 37 | 52 | 35 | 55 |
| temperature (°C) | 180 | 220 | 250 | 220 | 220 | 220 |
| time (min) | 30 | 30 | 30 | 30 | 60 | 60 |
| Eq. NaOPh | 2.2 | 2.2 | 2.2 | 4.4 | 4.4 | 3.3 |

critical effect on the yield. Temperatures of more than 200°C for half an hour give yields of no higher than 55%. Doubling the reaction time causes the yield to decrease drastically. An interpretation of this may be an alternative reaction path occurring. Under these reaction conditions, the starting material and a large excess of sodium phenoxide may lead to an irreversible formation of sodium chloride. The product itself may decompose to inorganic fluoride and unspecific fluorinated organic compounds.[1] It is recommended that the progress of the reaction is closely monitored for the maximal concentration of the diphenoxide and working the reaction up at that point. Besides qNMR, GC-MS is also a suitable method for monitoring. The nucleophilic substitution of the chlorine atom by the phenoxide ion in the sidechain of unsubstituted $p$-xylene proceeds via the well-known $S_N2$ pathway. This is not necessarily the case in the presence of a fluorinated side chain. Steric effects originating from the fluorine atoms lone pairs with the attacking nucleophile hinder the reaction. On the other hand, using a tosylate as the substrate may lead to high yields. However, nucleophilic substitution can proceed through a radical reaction via the $S_{RN}1$ mechanism. An essential feature of it is the reaction of the nucleophile with a fluorocarbon radical. Already electron-deficient substrate radicals would be made even more electrophilic due to the electron-withdrawing effect of the fluorine atoms. The reaction would be aided by UV irradiation. The fact, that some monophenoxide remains in the reaction mixture while all starting material has disappeared, is supposed to a sterically hindered attack of the phenoxide ion on the monophenoxy substitution product. If the phenoxide ion attacks the quinodimethane intermediate from only one side, the substitution may be hindered due to the interactions between the incoming bulky phenoxide ion and the phenoxy group, which is already in place. From a statistical point of view, a favourable course of the reaction would be facilitated by a simultaneous cleavage of the two C–Cl bonds, in order to form the less stabilized intermediate, which also plays a role in the CVD process (figure 4) [19]. Moreover, a particularly convenient feature of this reaction is the fact that phenol, which appears as a side-product during the subsequent polymerization reaction, could be recovered. After drying and recrystallization, it can be re-used as a reagent, which potentially makes it an economic synthesis.

---

[1]I exquisitely have to thank Prof. Lukas Hinterman from TU Munich for this information and Theresa Appleson for the kind supervision and assistance with NMR spectroscopy at the laboratories of the Catalysis Research Centre.

## 2. Experimental

This is the protocol for the thermal, laboratory-scale process: in a 250 ml three-necked round bottom flask equipped with a magnetic stirrer, internal thermometer, a silicone septum and a stopcock, 10 g (2.75 eq.) of anhydrous sodium hydride was suspended in 50 ml petrol ether under Ar atmosphere upon magnetic stirring. The stirrer was turned off and most of the supernatant petrol ether was decanted off. Through the silicone septum, 80 ml of DMPU was added with a syringe. Under Ar atmosphere, 19 g (2.2 eq.) of phenol was added portion-wise to the suspension, whereby a vigorous evolution of hydrogen gas ensued. Sixteen millilitres (23 g, 94 mmol) of 1,4-bis[difluoro(chloro)methyl]-benzene (ordered from ABCR/Germany) was added with a syringe through the silicone septum and the mixture was heated to reflux for 1 h. The colour of the mixture turned from light green to a deep dark green. The internal temperature was 182°C. The mixture was cooled to room temperature and poured into 400 ml of water upon vigorous magnetic stirring. A pink to purple suspension resulted. The pale-yellow precipitate was filtered and washed with 100 ml of water. After drying, 30 g of a light brown solid was obtained, which was subjected to vacuum distillation with a sickle condenser (180°C at 5 mbar). The obtained, light-yellow solid was recrystallized from boiling petrol ether to give 26 g (78% yield) of an off-white solid with an aromatic smell and a purity of greater than 99% according to the GC-MS data (mp: 124°C). To test whether varying parameters like excess of reagent, temperature and time influence the reaction, six small batches were prepared in the Anton Paar Monowave 300 microwave reactor in the Catalysis Research Centre of the Technical University in Munich in the research group of Prof. Dr Lukas Hintermann.

[1]H-NMR: (400 MHz, CDCl$_3$), $\delta$/ppm:
7.88 (4 H), 7.41 (dd, $J$ = 8.6 Hz, 4 H),
7.35–7.22 (m, 6 H)

[13]C-NMR: (101 MHz, CDCl$_3$), $\delta$/ppm:
150.32, 136.31 (t, $J$ = 32.4 Hz), 129.49,
125.96 (t, $J$ = 3.6 Hz), 125.80, 121.94,
121.67 (t, $J$ = 263 Hz)

[19]F-NMR: (377 MHz, CDCl$_3$), $\delta$/ppm:
−65.50

FT-IR: (KBr disk):
1323 cm$^{-1}$ (ss, C–F-stretching),
1201 cm$^{-1}$ (s, C–O-stretching),
849 cm$^{-1}$ (s, C–H-bending)
740 cm$^{-1}$ (s, C–H-bending)

This is the protocol for the microwave experiments: under Ar atmosphere, 2.2 (1.95 g) 3.3 eq. (2.87 g) and 4.4 eq. (3.81 g) of anhydrous sodium phenoxide were placed in microwave reaction vials, equipped with a magnetic stirrer. The solid was dissolved in 3 ml NMP per molar equivalent of sodium phenoxide. To each vial, 1.27 ml (1.87, 7.5 mmol) of 1,4-bis[difluoro(chloro)methyl]-benzene was added and placed in an Anton Paar Monowave 300. The vials were then heated. Temperature and reaction time of each individual batch are laid out in table 1. From each vial, The reaction mixture was poured into 200 ml water and extracted three times with a total volume of 150 ml diethyl ether. The combined organic phases were washed two times with a total volume of 100 ml water and dried over anhydrous sodium sulphate. After filtration and evaporation of the solvent in vacuo, the remaining solid material was taken up in chloroform and 250 μl of trichloroethylene was added. The resulting solution was used for qNMR studies. Corresponding results are shown in table 1.

**Starting material**
[1]H-NMR: (400 MHz, CDCl$_3$), $\delta$/ppm:
7.74 (4 H),
[19]F-NMR: (377 MHz, CDCl$_3$), $\delta$/ppm:
−49.87

### Mono substitution product

$^1$H-NMR: (400 MHz, CDCl$_3$), $\delta$/ppm:
7.87–783 (d,m $J$ = 8.2, 2 H),
7.78–7.72 (d,m $J$ = 8.2 Hz, 2 H),
7.35–7.22 (m, 6 H)
$^{19}$F-NMR: (377 MHz, CDCl$_3$), $\delta$/ppm:
−49.55, −65.61

### Diphenoxide

$^1$H-NMR: (400 MHz, CDCl$_3$), $\delta$/ppm:
7.84 (4 H), 7.35–7.22 (m, 6 H)
$^{19}$F-NMR: (377 MHz, CDCl$_3$), $\delta$/ppm:
−65.49

Data accessibility. Spectroscopic data (GC-MS, IR, NMR) of the diphenoxide product is available in the Dryad Digital Repository: https://doi.org/10.5061/dryad.tx95x69wc [20].

Authors' contributions. G.F. is the supervisor of this project and cleared the necessary facilities to do the work. J.J.S. provided useful information about the substance to be synthesized and contributed some great ideas for the work.

Competing interests. We declare we have no competing interests.

Funding. The work was funded by the Federal Ministry of Education and Research (grant no. 03INT501AD). To meet the publication costs, financial support of the Federal Secretary of Economy under project number KF257-5103 CR4 is gratefully appreciated.

Acknowledgements. The preparative work has been carried out in the laboratories of Plasma Parylene Services in Rosenheim, Germany. Special thanks are due to Dieter Voss as owner of this company. The analytical work was performed at the Ludwig-Maximilian-University Munich, Department of Chemistry under the supervision of Claudia Ober (NMR) and Werner Spahl (GC-MS) as well as at the Technical University Munich, Department of Chemistry under the supervision of Theresa Appleson and Lukas Hintermann (qNMR and microwave-assisted synthesis). Special thanks go out to Michael Cappi at Aroma-Lab AG. Without their help, this work would have not been possible.

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
