## [Peer Review File · Royal Society Open Science]

Review History

RSOS-201921.R0 (Original submission)

Review form: Reviewer 1

Is the manuscript scientifically sound in its present form?

Yes

Are the interpretations and conclusions justified by the results?

Yes

Is the language acceptable?

Yes

Do you have any ethical concerns with this paper?

No

Have you any concerns about statistical analyses in this paper?

No

Recommendation?

Accept with minor revision (please list in comments)

Comments to the Author(s)

The authors reported a new method to synthesize parylene AF-4 precursors. They reported a yield as high as 78%. The work is meaningful. A few suggestions were included as the purity of the precursor and demonstration of CVD applicability.

Review form: Reviewer 2

Is the manuscript scientifically sound in its present form?

Yes

Are the interpretations and conclusions justified by the results?

Yes

Is the language acceptable?

Yes

Do you have any ethical concerns with this paper?

No

Have you any concerns about statistical analyses in this paper?

No

Recommendation?

Major revision is needed (please make suggestions in comments)

Comments to the Author(s)

Comment to RSOS-201921

In this work, Kast et al. reported the synthesis of an alternative precursor for CVD of parylene AF-4. The suggestions are as follows.

- (1) Figure 1 should be noted in the manuscript.
- (2) As shown in the Table 1, the Dimer is 55%, so the other components should be clarified.
- (3) The purity of compound (3) should be noted.
- (4) Please discuss the advantages of the method in this work compared to previous work.
- (6) The language should be improved, and there are several mistakes and errors.

Decision letter (RSOS-201921.R0)

Dear Mr Kast:

Title: Improved route to a diphenoxide-based precursor for CVD of parylene AF-4
Manuscript ID: RSOS-201921

Thank you for submitting the above manuscript to Royal Society Open Science. On behalf of the Editors and the Royal Society of Chemistry, I am pleased to inform you that your manuscript will be accepted for publication in Royal Society Open Science subject to minor revision in accordance with the referee suggestions. Please find the reviewers' comments at the end of this email.

The reviewers and handling editors have recommended publication, but also suggest some minor revisions to your manuscript. Therefore, I invite you to respond to the comments and revise your manuscript.

Because the schedule for publication is very tight, it is a condition of publication that you submit the revised version of your manuscript before 28-Feb-2021. Please note that the revision deadline will expire at 00.00am on this date. If you do not think you will be able to meet this date please let me know immediately.

Once again, thank you for submitting your manuscript to Royal Society Open Science. The chemistry content of Royal Society Open Science is published in collaboration with the Royal

Society of Chemistry. I look forward to receiving your revision. If you have any questions at all, please do not hesitate to get in touch.

Kind regards,
Dr Laura Smith
Publishing Editor, Journals

On behalf of the Subject Editor Professor Anthony Stace and the Associate Editor Dr Andrew Harned.

RSC Associate Editor:

Comments to the Author:

This is a focused manuscript that describes the preparation of a new precursor for certain CVD experiments. I am of the opinion that this work will prove useful to others in the area and will command some attention. The reviewers have noted a few relatively minor items that should be addressed by the authors. In particular, demonstrating purity of the final diphenoxide product is important given the started purpose of this compound.

One reviewer suggests that the authors demonstrate that the target compound can actually be used in CVD experiments. This would be a nice addition to strengthen the manuscript, but I am willing to let the authors defer this demonstration if they feel that is better suited to a follow up paper

In addition to the concerns raised by the referees, I have a few additional items:

- (1) Please include copies of the NMR spectra (^1H , ^{13}C , and ^{19}F) as electronic supplementary materials
- (2) please modify the manuscript so that compound identification is consistent. In some places Roman numerals are used. In others, letters (E.g. A, B, C) are used. It would also be helpful if the identifiers are added to the figures and Table as appropriate.
- (3) Please ensure that the spectral data in the experimental section are listed in such a way that all the data for each compound is together. In the current state, some data is broken up and this could lead to confusion by future readers.

RSC Subject Editor:

Comments to the Author:

(There are no comments.)

Reviewer comments to Author:

Reviewer: 1

Comments to the Author(s)

The authors reported a new method to synthesize parylene AF-4 precursors. They reported a yield as high as 78%. The work is meaningful. A few suggestions were included as the purity of the precursor and demonstration of CVD applicability.

Reviewer: 2

Comments to the Author(s)

Comment to RSOS-201921

In this work, Kast et al. reported the synthesis of an alternative precursor for CVD of parylene AF-4. The suggestions are as follows.

- (1) Figure 1 should be noted in the manuscript.
- (2) As shown in the Table 1, the Dimer is 55%, so the other components should be clarified.
- (3) The purity of compound (3) should be noted.
- (4) Please discuss the advantages of the method in this work compared to previous work.
- (6) The language should be improved, and there are several mistakes and errors.

Author's Response to Decision Letter for (RSOS-201921.R0)

See Appendix A.

Decision letter (RSOS-201921.R1)

Dear Professor Franz:

Title: Improved route to a diphenoxide-based precursor for CVD of parylene AF-4
Manuscript ID: RSOS-201921.R1

It is a pleasure to accept your manuscript in its current form for publication in Royal Society Open Science. The chemistry content of Royal Society Open Science is published in collaboration with the Royal Society of Chemistry.

On behalf of the Subject Editor Professor Anthony Stace and the Associate Editor Dr Andrew Harned.

RSC Associate Editor
Comments to the Author:
(There are no comments.)

Reviewer(s)' Comments to Author:

Appendix A

Munich, 23rd February 2021

Dear Dr. Smith,

we are speechless and overwhelmed, thank you very much! I will now address the suggestions of the reviewers in detail:

Regarding your comments:

1. The spectroscopic data has been uploaded to my Dryad repository. The data set is already published and can be accessed via <https://doi.org/10.5061/dryad.tx95x69wc>. Spectra as PDFs, as well as files in .jdx-format of the NMR and infrared analyses are included. Unfortunately, the GC-MS data are only available as PDF, since the analytics supervisor at the LMU in Munich missed out to submit the raw data to me via email, and the files are probably already deleted. I talked about this issue before with one of your colleagues who said that it is fine that way.
2. The compound identification has been revised and embedded in the manuscript accordingly.
3. There is only one molecule/compound addressed in the manuscript. All spectra except for the qNMR dataset are intended to demonstrate the purity and chemical identity of the diphenoxide.

Regarding the comments of reviewer 1:

Since no CVD experiments were undertaken in this work, we considered to focus just on the recipe to obtain the precursor and on the chemical description of its properties. Detailed information about the applicability of the diphenoxide for CVD is presented in Jay. J. Senkevichs paper “Parylene AF-4 via the Trapping of a Phenoxy Leaving Group” published in the Journal of Chemical Vapor Deposition in 2013 (DOI: 10.1002/cvde.201304321), which is now referenced in the text.

Regarding the comments of reviewer 2:

1. Figure 1 is now noted in the text in the following passage:

“From a statistical point of view, a favourable course of the reaction would be facilitated by a simultaneous cleavage of the two C-Cl bonds, in order to form the less stabilized intermediate, which also plays a role in the CVD process (see fig. 1).”

2. In table one, the entries “Di” and “Mono” have been changed to “Diphenoxide” and “Monophenoxide”. This should clarify the compound which has been subjected to qNMR experiments.
3. The purity of the final product is now noted together with its melting point at the end of the section of the thermal experiments.

4. The results of previous works were addressed in the following passage:

“Earlier protocols like the one of Dolbier et al³ used highly toxic HMPA. We decided to replace it by DMPU, which poses a far smaller safety hazard and environmental concerns, which poses a very important consideration for scaleup.¹⁵ The solvent HMPA is known to favour S_N2 reactions and to increase their rate. How exactly the mechanism works is though not clear amongst all authors of previous related publications.³^{16,17} In microwave reactions and subsequent qNMR studies, the influence of temperature, time and excess molar equivalents of sodium phenolate were compared. The results are listed in table 1. Apparently, high temperatures above 150 °C are needed to achieve a considerable conversion of the dichloride to the diphenoxide. The reaction temperature in the experiments of DOLBIER et al. and GUIDOTTI et al. was between 100 °C and 120 °C.”

5. The text was checked for correct spelling and language. Mr. Senkevich as a native english speaker confirmed this.

Please don't hesitate to get back in touch with us, if there are still points that are unclear.

Yours sincerely,

Daniel Kast